# MADGen:Minority Attribute Discovery in Text-to-Image Generative Models

## Abstract

Text-to-image diffusion models achieve impressive generation quality but also inherit and amplify biases from training data, resulting in biased coverage of semantic attributes. Prior work addresses this in two ways. Closed-set approaches mitigate biases in predefined fairness categories (e.g., gender, race), assuming socially salient minority attributes are known a priori. Open-set approaches frame the task as bias identification, highlighting majority attributes that dominate outputs. Both overlook a complementary task: uncovering minority features underrepresented in the data distribution (social, cultural, or stylistic) yet still encoded in model representations. We introduce MADGen, the first framework, to our knowledge, for discovering minority attributes in diffusion models. Our method leverages Matryoshka Sparse Autoencoders and introduces a minority metric that integrates neuron activation frequency with semantic distinctiveness, enabling the unsupervised identification of rare attributes. Specifically, MADGen identifies a set of neurons whose behavior can be directly interpreted through their top-activating images, which correspond to underrepresented semantic attributes in the model. Quantitative and qualitative experiments demonstrate that MADGen uncovers attributes beyond fairness categories.

## 1 Introduction

Text-to-image (T2I) diffusion models such as Stable Diffusion have revolutionized image generation by producing high-fidelity visuals from natural language prompts (Podell et al., 2023; Rombach et al., 2022). However, these models not only reflect biases from their training data (Luccioni et al., 2023; Perera and Patel, 2023), but can also amplify them during generation, reinforcing societal stereotypes and inequalities if left unaddressed (Seshadri et al., 2024). Such disparities reduce semantic coverage and raise concerns about fairness, representation, and deployment in real-world settings.

Several approaches counteract biases in T2I generative models by rebalancing or diversifying their outputs (Chuang et al., 2023; Ni et al., 2023; Shen et al., 2024; Li et al., 2024). While effective for predefined categories like gender or race, they often overlook subtler underrepresentation such as physical traits, cultural symbols, or stylistic variations, essential for semantic diversity and faithful generation. Open-set bias detection (D'Incà et al., 2024) broadens auditing but mainly identifies the majority attributes and the surfaced attributes are largely dictated by inductive biases of external world models. Suppressing such majority features does not amplify the underrepresented ones, overlooking the critical task of identifying *minority attributes*: semantic factors encoded in the model's internal representations, but consistently underexpressed. Auditing remains incomplete without discovering them: it reveals what the model overproduces, but not what it neglects. Moreover, discovering minority attributes is vital for diversity and representation, counterbalancing imbalances and capturing cultural, stylistic nuances beyond fairness axes essential for realism and expressivity.

Submitted to 39th Conference on Neural Information Processing Systems (NeurIPS 2025). Do not distribute.

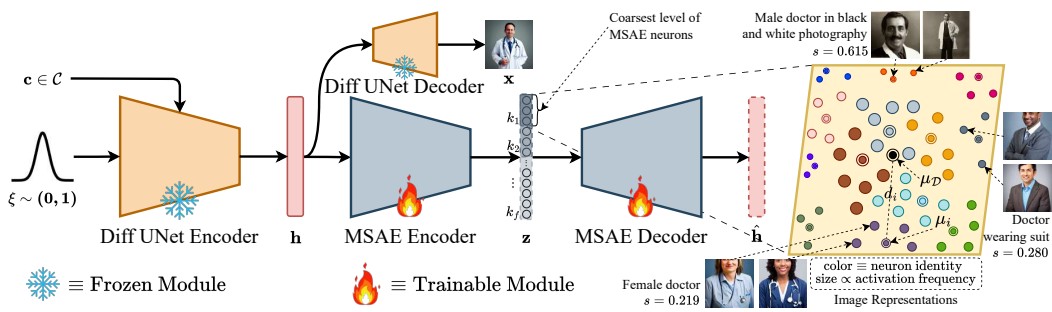

Figure 1: Overview of **MADGen**. Diffusion representations (**h**) are decomposed by MSAE into interpretable features (**z**). A minority score ($s$), combining rarity and distinctiveness, ranks neurons or features to reveal *minority attributes*. Minority concepts are identified at the coarsest MSAE level (e.g., female doctor, doctor in suit), where size reflects activation frequency (smaller size = less frequent) and color denotes neuron identity.

To address this gap, we introduce **MADGen**, the first framework for **Minority Attribute Discovery** in diffusion models without reliance on external language models. Rather than identifying all possible underrepresented attributes, MADGen targets those that are already encoded in the internal representations of the model but are systematically underexpressed during generation. These attributes are not hallucinated or externally defined, but emerge directly from the learned feature space of the model. By shifting the focus from majority identification to the structured discovery of these suppressed features, MADGen enables more comprehensive auditing and a deeper understanding of representational gaps in generative models.

Identifying minority attributes requires access to the internal factors of variation learned by diffusion models. However, these representations are often entangled and uninterpretable. We adopt Matryoshka Sparse Autoencoders (MSAEs), which have shown strong interpretability in vision-language models via hierarchical semantic decomposition (Pach et al., 2025; Zaigrajew et al., 2025). We apply MSAE to diffusion activations and focus on the coarsest level, where neurons capture broad, high-level semantics. In contrast, finer layers often fragment concepts into narrow or stylistic variations, making them less reliable for discovering underrepresented attributes. Among coarse neurons, we identify minority-associated ones using a metric combining: (i) activation frequency, measuring how rarely a neuron fires, and (ii) semantic distinctiveness, measuring how dissimilar its top-activating samples are from the dataset distribution. This highlights neurons encoding rare yet semantically coherent concepts systematically underrepresented in outputs. Importantly, the interpretable structure of MSAEs enables direct visual interpretation through top-activating examples and spatial heatmaps, making explicit the attributes each neuron captures.

The key contributions of this work are as follows: ❶ To the best of our knowledge, we introduce the first framework for minority attribute discovery in diffusion models, extending bias analysis from predefined fairness categories or majority-dominant features to the systematic identification of underrepresented attributes encoded in model representations. ❷ We propose a simple, yet effective, minority metric that combines neuron activation frequency with semantic distinctiveness, forming the basis of MADGen. ❸ We present both quantitative and qualitative evidence that MADGen uncovers attributes extending beyond standard fairness categories.

## 2 Methodology

We propose **MADGen**, a framework for minority attribute discovery in text-to-image diffusion models. An overview of the framework is illustrated in Figure 1.

### 2.1 Minority Attribute Discovery

**Feature Decomposition from Diffusion Representations:** To expose the internal concepts encoded by diffusion models, we train MSAE on intermediate representations extracted during reverse sampling. Given a T2I diffusion model $G$ and a prompt **c**, we extract bottleneck representations

72 $\mathbf{h}_t \in \mathbb{R}^{h \times w \times n}$ at each denoising step $t$. These representations are inherently interpretable (Kwon
73 et al., 2023), and Kim et al. (2025) has shown that SAEs trained on them reveal high-level features.
74 Following Cywiński and Deja (2025), we treat each spatial location in $\mathbf{h}_t$ as an $n$-dimensional training
75 example, disregarding spatial coordinates. These vectors are then used to train a MSAE, yielding a
76 hierarchy of sparse codes $\mathbf{z}^{(k_i)}$ that capture semantic structure at varying levels of granularity from
77 broad concepts at coarse levels $(k_1)$ to finer details at deeper levels $(k_f)$.

78 For minority attribute discovery, we perform inference with MSAE by collecting representation-image
79 pairs $\mathcal{D}_c = \{ (\mathbf{h}_t^{(j)}, \mathbf{x}^{(j)}) \}_{j=1}^N$, where $\mathbf{h}_t^{(j)} \in \mathbb{R}^{h \times w \times n}$ denotes bottleneck representation at a fixed
80 denoising step $t$, and $\mathbf{x}^{(j)}$ is the corresponding generated image for a prompt $c$. In practice, we use
81 the final timestep, where semantic information is most fully expressed. For simplicity, we omit both
82 the sample index $j$ and the timestep index $t$, and write $(\mathbf{h}, \mathbf{x}) \in \mathcal{D}_c$ for an arbitrary pair. Each tensor
83 $\mathbf{h}$ is flattened into $h \times w$ feature vectors, which are individually passed through the MSAE encoder
84 following the training setup. For each MSAE neuron $z_i$, with $i = 1, \ldots, d$ corresponding to a sparse
85 latent feature, we define its activation on $\mathbf{h}$ as $z_i(\mathbf{h})$, obtained by averaging activations across spatial
86 positions. These per-neuron activations form the basis for computing the minority score.

87 **Minority Score:** To quantify the degree to which each neuron encodes a minority attribute, we
88 introduce the *Minority Score*, which balances two complementary signals: rarity of activation and
89 semantic distinctiveness. Let $(\mathbf{h}, \mathbf{x}) \in \mathcal{D}_c$ be an diffusion representation-image pair, and $z_i(\mathbf{h})$ the
90 activation of MSAE neuron $z_i$ as previously defined. We define the activation frequency as the
91 proportion of samples where the neuron is active (i.e., has nonzero activation):

$$\nu_i = \frac{|\{(\mathbf{h}, \mathbf{x}) \in \mathcal{D}_c : z_i(\mathbf{h}) > 0\}|}{|\mathcal{D}_c|}. \tag{1}$$

92 This metric directly measures how often the neuron participates across the dataset $\mathcal{D}_c$, with rarer
93 features corresponding to lower $\nu_\mathbf{i}$. While activation frequency identifies neurons that fire rarely,
94 rarity alone is insufficient: some neurons may activate sparsely but correspond to noisy, low-level
95 fluctuations that are not meaningful for interpretation. To address this, we evaluate the semantic
96 distinctiveness of each neuron by comparing its activation-weighted CLIP centroid to the global
97 dataset centroid. Let $\mathrm{CLIP}(\mathbf{x})$ denote the CLIP embedding of image $\mathbf{x}$. The centroid $\mu_i$ for neuron
98 $z_i$, and the global centroid $\mu_{\mathcal{D}_c}$, are computed as:

$$\mu_i = \frac{\sum_{(\mathbf{h}, \mathbf{x}) \in \mathcal{D}_c} z_i(\mathbf{h}) \mathrm{CLIP}(\mathbf{x})}{\sum_{(\mathbf{h}, \mathbf{x}) \in \mathcal{D}_c} z_i(\mathbf{h})}, \qquad \mu_{\mathcal{D}_c} = \frac{1}{|\mathcal{D}_c|} \sum_{(\mathbf{h}, \mathbf{x}) \in \mathcal{D}_c} \mathrm{CLIP}(\mathbf{x}). \tag{2}$$

99 Semantic distinctiveness $d_i$ is then defined as the cosine distance between the two centroids. This
100 metric ensures that the neuron centroid $\mu_i$ is dominated by its top-activating images, since activation
101 strengths directly weight their contribution. In contrast, the global dataset centroid $\mu_{\mathcal{D}_c}$ represents
102 the average semantics of the entire set of images in $\mathcal{D}_c$, dominated by the majority distribution.
103 The resulting distance $d_i$ thus measures how much the concept encoded by a neuron diverges from
104 dominant patterns in the data. Both $d_i$ and $\nu_i$ are min–max normalized to $[0, 1]$ for comparability.
105 Finally, *Minority Score* is defined as:

$$s(\mathbf{z}) = \mathbf{d} \odot (\mathbf{1} - \boldsymbol{\nu}) \tag{3}$$

106 where $\mathbf{d} = (d_1, \ldots, d_d)$, $\boldsymbol{\nu} = (\nu_1, \ldots, \nu_d)$. This formulation assigns a high score to neurons that are
107 both rarely active (low $\nu_i$) and semantically distinct (high $d_i$) relative to the majority distribution.
108 Intuitively, neurons with larger $s(z_i)$ are more likely to encode minority attributes, since they capture
109 concepts that occur infrequently yet deviate substantially from dominant patterns. Conversely,
110 neurons with the lowest minority scores do not necessarily correspond to dominant attributes, since
111 low values can also arise from noisy or undistinctive activations. Although the Minority Score can
112 be computed across all MSAE neurons, we focus on the coarsest level $z(k_1)$, which captures broad,
113 interpretable semantics. By restricting analysis to the top-$k_1$ codes, we prioritize high-level structure
114 over low-level noise, leveraging the hierarchical design of MSAEs to expose global attributes.

115 Minority concepts often appear redundantly across multiple neurons with similar activation patterns.
116 To obtain a compact and diverse set, we use the neuron centroid $\mu_i$ (Eq. 2) as a representative of
117 each neuron's semantics. Redundancy is assessed via pairwise cosine distances between centroids,
118 which quantify similarity between neurons. We then iterate over neurons in descending order of
119 *Minority Score*, retaining the current neuron in the final set and removing all others within a small

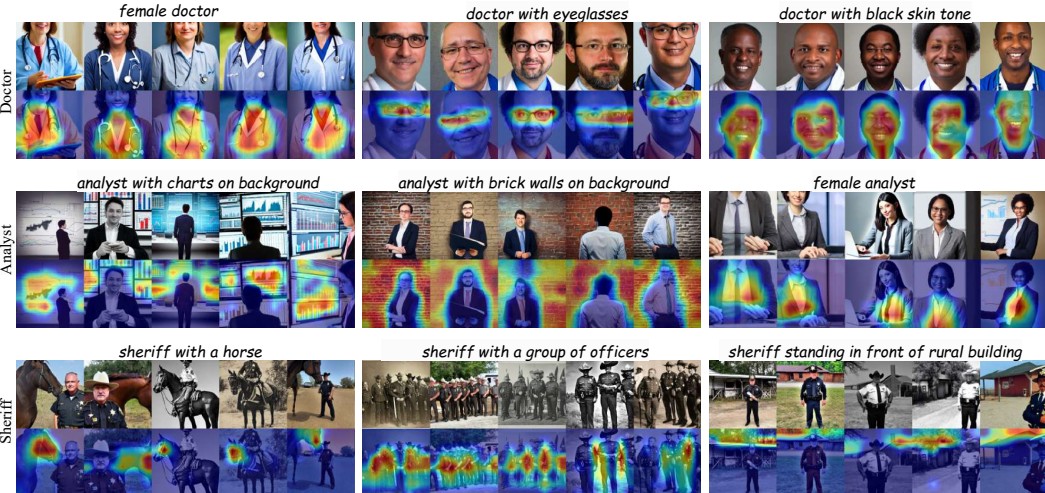

Figure 2: Top-activating images and MSAE activation heatmaps for minority neurons discovered by MADGen across three WinoBias prompts. **Top:** "Doctor." **Middle:** "Analyst." **Bottom:** "Sheriff." The label above each image shows its language-based annotation.

fixed distance. This threshold, treated as a hyperparameter, controls semantic redundancy. To further ensure that the final set contains minority neurons, we restrict analysis to those above the 90th percentile of the score distribution. For interpretability, we visualize top-activating images with MSAE heatmaps and provide human-readable annotations via LLMs.

## 3 Experiments

We conduct qualitative and quantitative evaluations of the minority attributes discovered by MADGen.

### 3.1 Minority Attributes Discovery

**Experimental setting:** We perform minority attribute discovery on Stable Diffusion v1.4 (Rombach et al., 2022) on WinoBias (Zhao et al., 2018) and COCO prompts (Lin et al., 2015) to ground our analysis in datasets that capture complementary aspects of bias and diversity. For each prompt, we generate images, capture bottleneck activations to train an MSAE, then identify minority attributes (Sec. 2) and obtain human-readable annotations with GPT-4o.

We assess the effectiveness of MADGen using two complementary metrics. ❶**Likelihood:** This metric quantifies how probable the discovered minority attributes are under the generative model's distribution (Song et al., 2020). For each neuron identified as encoding a minority attribute, we evaluate the likelihoods of its top-activating images and compare them against randomly sampled images from SD under the same prompts. Minority attributes are expected to lie in lower-density regions of the distribution, resulting in lower likelihood values. ❷ **Attribute presence:** This metric measures how often the minority attributes discovered by MADGen appear in images generated from SD. We use the language annotations obtained for these attributes as described in Section 2, and for each generated image, evaluate whether the annotated attribute is detected by the LLaMA-4 Scout model. Lower presence values indicate stronger underrepresentation. We compare MADGen against OpenBias (D'Incà et al., 2024), which targets majority attributes. As no prior method addresses minority attribute discovery, this provides a complementary perspective: while open-set methods expose what the model overproduces, MADGen uncovers what it systematically suppresses.

**Results:** We evaluate MADGen using average likelihood and attribute presence metrics to assess its ability to uncover minority attributes in diffusion models. Table 1 shows that attributes linked to MADGen's minority neurons yield significantly lower likelihoods than random Stable Diffusion samples, indicating that these features occupy low-density regions, encoded in the representation space but rarely expressed in generated images—supporting our definition of minority attributes as underrepresented yet nonzero. Table 2 further shows that OpenBias surfaces attributes with

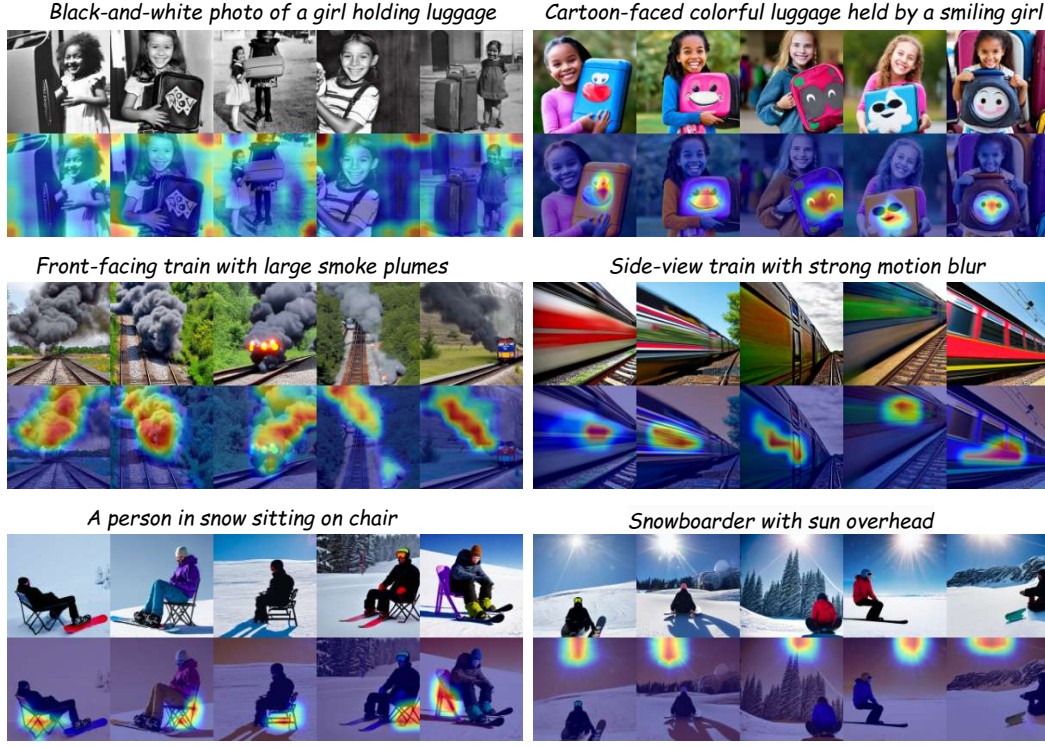

Figure 3: Top-activating images and MSAE activation heatmaps for minority neurons discovered by MADGen across COCO prompts. **Top:** "A girl smiling as she holds a piece of luggage." **Middle:** "A train going down a track at full speed." **Bottom:** "A person sitting in the snow with snowboard on." The label above each image shows its language-based annotation.

high frequency, while those revealed by MADGen appear far less often, providing direct evidence that MADGen isolates rare modes complementary to the dominant ones captured by OpenBias. Qualitative results in Fig. 2 and Fig.3 show that MADGen reveals both socially salient minority attributes (e.g., female, black skin tone), non-fairness concepts (e.g., analysts with brick walls, sheriffs in rural settings), and rare stylistic/contextual features (e.g., black-and-white portraits, side-view trains with motion blur), demonstrating its ability to generalize beyond fairness categories and capture diverse underrepresented semantics.

Table 1: Average likelihood measured in bits per dimension.

| Approach | WinoBias ($\downarrow$) | COCO ($\downarrow$) |
|---|---|---|
| Stable Diffusion | 2.394 | 2.378 |
| MADGen | **2.371** | **2.334** |

Table 2: Attribute presence for majority (Open-Bias) and minority (MADGen).

| Approach | WinoBias ($\downarrow$) | COCO ($\downarrow$) |
|---|---|---|
| OpenBias | 0.941 | 0.933 |
| MADGen | **0.256** | **0.265** |

## 4 Conclusions

We propose MADGen, a framework for minority attribute discovery in diffusion models that combines Matryoshka Sparse Autoencoders with a novel minority score to identify features encoded in latent representations but underrepresented at generation. Unlike prior work focused on fixed fairness categories or majority trends, MADGen uncovers rare, semantically meaningful attributes directly from internal activations. Through quantitative and qualitative analyses, we show that MADGen reveals fairness, stylistic, and cultural minorities. By grounding discovery in model representations, MADGen complements LLM-based bias tools and lays the foundation for hybrid auditing frameworks that bridge external priors with internal model dynamics.

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
