# OpenReview forum: "MADGen:Minority Attribute Discovery in Text-to-Image Generative Models"
_EurIPS.cc/2025/Workshop/UPLB — UPLB2025_

### Official Review · Reviewer_GJQR · 2025-10-25
**A tool to identify features encoded in the internal representations of text-to-image diffusion models but underrepresented in generation**

**Rating:** 7
**Confidence:** 3

**Review:**

This paper introduces a novel metric aimed at identifying minority attributes encoded within trained models. The method leverages a Multiscale Sparse Autoencoder (MSAE), which enables the extraction of interpretable features in a hierarchical manner. The proposed metric identifies neurons that are rarely activated yet correspond to semantically coherent concepts that are systematically underrepresented in the model’s outputs.

The tool appears useful for revealing underrepresented features in a human-interpretable way, as it can hierarchically highlight the regions of generated images most relevant to those features—insights that could, in principle, inform bias mitigation strategies during dataset design. However, the paper is challenging to follow, and it remains unclear how the proposed method can be concretely applied to mitigate generation bias. The figures do not substantially clarify the results, as the activation heatmaps seem to emphasize the most informative features related to the textual input, and the connection with underrepresentation is not clearly explained. Additionally, Table I is poorly explained and difficult to interpret.

Overall, the metric is interesting and conceptually well-motivated, and the objectives of the paper align well with the workshop’s focus. The proposed tool has the potential to become a valuable instrument for analyzing diffusion models. However, the presentation would benefit from greater clarity, the inclusion of a more rigorous statistical analysis, stronger empirical validation, and a clearer discussion of the broader perspectives and implications opened by this work.

---

### Decision · Program_Chairs · 2025-11-03

Accept (Poster)